# Focal Modulation Networks

**Jianwei Yang, Chunyuan Li, Xiyang Dai, Jianfeng Gao**
{jianwyan,chunyl,xidai,jfgao}@microsoft.com

## Abstract

We propose *focal modulation networks* (*FocalNets* in short), where self-attention (SA) is completely replaced by a *focal modulation* module for modeling token interactions in vision. Focal modulation comprises three components: ($i$) hierarchical contextualization, implemented using a stack of depth-wise convolutional layers, to encode visual contexts from short to long ranges, ($ii$) gated aggregation to selectively gather contexts for each query token based on its content, and ($iii$) element-wise modulation or affine transformation to fuse the aggregated context into the query. Extensive experiments show FocalNets outperform the state-of-the-art SA counterparts (*e.g.*, Swin and Focal Transformers) with similar computational cost on the tasks of image classification, object detection, and semantic segmentation. Specifically, FocalNets with tiny and base size achieve **82.3**% and **83.9**% top-1 accuracy on ImageNet-1K. After pretrained on ImageNet-22K, it attains **86.5**% and **87.3**% top-1 accuracy when finetuned with resolution $224^2$ and $384^2$, respectively. When transferred to downstream tasks, FocalNets exhibit clear superiority. For object detection with Mask R-CNN, FocalNet base trained with 1$\times$ outperforms the Swin counterpart by **2.1** points and already surpasses Swin trained with 3$\times$ schedule (**49.0** *v.s.* 48.5). For semantic segmentation with UPerNet, FocalNet base at single-scale outperforms Swin by **2.4**, and beats Swin at multi-scale (**50.5** *v.s.* 49.7). Using large FocalNet and mask2former, we achieve **58.5** mIoU for ADE20K semantic segmentation, and **57.9** PQ for COCO Panoptic Segmentation. These results render focal modulation a favorable alternative to SA for effective and efficient visual modeling. Code is available at: https://github.com/microsoft/FocalNet.

## 1 Introduction

Transformers [66], originally proposed for natural language processing (NLP), have become a prevalent architecture in computer vision since the seminal work of Vision Transformer (ViT) [19]. Its promise has been demonstrated in various vision tasks including image classification [63, 69, 74, 46, 87, 65], object detection [3, 97, 93, 15], segmentation [67, 72, 13], and beyond [38, 91, 4, 9, 68, 36]. In Transformers, the self-attention (SA) is arguably the key to its success which enables input-dependent global interactions, in contrast to convolution operation which constrains interactions in a local region with a shared kernel. Despite this advantages, the efficiency of SA has been a concern due to its quadratic complexity over the number of visual tokens, especially for high-resolution inputs. To address this, many works have proposed SA variants through token coarsening [69], window attention [46, 65, 87], dynamic token selection [51, 81, 50], or the hybrid [79, 14]. Meanwhile, a number of models have been proposed by augmenting SA with (depth-wise) convolutions to capture long-range dependencies with a good awareness of local structures [74, 22, 78, 20, 18, 35, 7, 17].

In this work, we aim at answering the fundamental question: *Is there a better way than (hybrid) SA to model input-dependent long-range interactions?* We start with an analysis on the current advanced designs for SA. In Fig. 1(a), we show a window-wise attention between the red query token and the surrounding orange tokens proposed in Swin Transformer [46]. With a simple window-shift

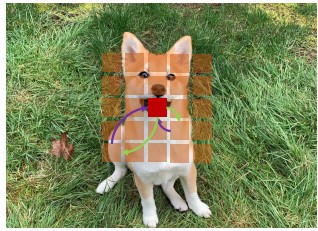
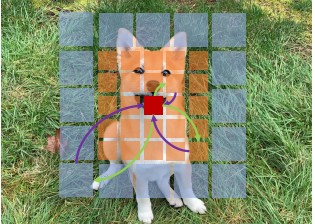
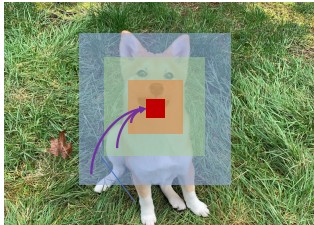

(a) Window-wise SA       (b) Focal Attention       (c) Focal Modulation

Figure 1: Illustrative comparison among (a) Window-wise Self-Attention (SA) [46], (b) Focal Attention (FA) [79] and (c) the proposed Focal Modulation. Given the query token ▮, window-wise SA captures spatial context from its surrounding tokens ▮, FA additionally uses far-away summarized tokens ▮, and Focal Modulation first encodes spatial context at different levels of granularity into summarized tokens ( ▮ , ▮ , ▮ ), which are then adaptively fused into the query token depending on the query content. Green and purple arrows represent the attention interactions and query-dependent aggregations, respectively (we do not draw all arrows for clarity). Both window-wise self-attention and focal attention involve heavy interaction and aggregation operations, while our focal modulation turn both of them light-weight. Figures better viewed in color.

strategy, Swin attains superior performance to ResNets across various vision tasks. To enlarge the receptive field, focal attention [79] is proposed to additionally aggregate summarized visual tokens far away to capture coarse-grained, long-range visual dependencies, as shown in Fig. 1(b). To produce the outputs, both methods involve heavy interactions (green arrows) followed by equally heavy aggregations (purple arrows) between the query and a large number of spatially distributed tokens (context features), which are extracted via either window partition or unfolding. In this work, we take an alternative way by *first aggregating contexts around each query and then modulating the query with the aggregated context*. This alteration still enables input-dependent token interaction, but significantly eases the process by decoupling the aggregation from individual queries, hence making the interactions light-weight upon a couple of features. As shown in Fig. 1(c), we can simply apply query-agnostic aggregations (*e.g.*, depth-wise convolution) to generate summarized tokens at different levels of granularity. Afterwards, these summarized contexts are selectively aggregated depending on the query content, and finally fused into the query vector. We call this new token interaction mechanism *focal modulation*, with which we replace SA in Transformers to build a simpler and attention-free architecture, called *Focal Modulation Network*, or *FocalNet* in short.

Finally, extensive experiments on image classification, object detection and segmentation, show that our FocalNets consistently and significantly outperform the SoTA SA counterparts with comparable costs. Notably, our FocalNet achieves **82.3%** and **83.9%** top-1 accuracy using tiny and base model size, but with comparable and doubled throughput than Swin and Focal Transformer, respectively. When pretrained on ImageNet-22K, our FocalNets achieve **86.5%** and **87.3%** in $224^2$ and $384^2$ resolution, respectively, which are comparable or better than Swin at similar cost. The advantage is particularly significant when transferred to dense prediction tasks. For object detection on COCO [42], our FocalNets with tiny and base model size achieve **46.1** and **49.0** box mAP on Mask R-CNN $1\times$, surpassing Swin with $3\times$ schedule (46.0 and 48.5 box mAP). For semantic segmentation on ADE20k [95], our FocalNet with base model size achieves **50.5** mIoU at single-scale evaluation, outperforming Swin at multi-scale evaluation (49.7 mIoU). Using the pretrained large FocalNet, we achieve **58.5** mIoU for ADE20K semantic segmentation, and **57.9** PQ for COCO Panoptic Segmentation based on Mask2former [12]. Furthermore, we apply our focal modulation to monolithic ViT and clearly demonstrate superior performance across different model sizes.

## 2 Related Work

**Self-attentions**. Transformer [66] is first introduced to vision in Vision Transformer (ViT) [19] by splitting an image into a sequence of visual tokens. The self-attention (SA) strategy in ViTs has demonstrated superior performance to modern convolutional neural networks (ConvNets) such as ResNet [27] when trained with optimized recipes [19, 63]. Afterwards, multi-scale architectures [5, 69, 78], light-weight convolution layers [74, 22, 39], local self-attention mechanisms [46, 87, 14, 79] and learnable attention weights [84] have been proposed to boost the performance and support high-resolution input. More comprehensive surveys are covered in [34, 24, 34]. Our focal modulation significantly differs from SA by first aggregating the contexts from different levels of granularity and then modulating individual query tokens, rendering an attention-free mechanism for token

interactions. For context aggregation, our method is inspired by focal attention proposed in [79]. However, the context aggregation for focal modulation is performed at each query location instead of target location, followed by a modulation rather than an attention. These differences in mechanism lead to significant improvement of efficiency and performance as well. Another closely related work is Poolformer [83] which uses a pooling to summarize the local context and a simple subtraction to adjust the individual inputs. Though achieving decent efficiency, Poolformer lags behind popular vision transformers like Swin on performance. As we will show later, capturing local structures at different levels is essential for performance.

**MLP architectures**. Visual MLPs can be categorized into two groups: ($i$) Global-mixing MLPs, such as MLP-Mixer [60] and ResMLP [62], perform global communication among visual tokens through spatial-wise projections augmented by various techniques, such as gating, routing, and Fourier transforms [44, 49, 58, 59]. ($ii$) Local-mixing MLPs sample nearby tokens for interactions, using spatial shifting, permutation, and pseudo-kernel mixing [82, 29, 41, 8, 23]. Recently, Mix-Shift-MLP [92] exploits both local and global interactions with MLPs, in a similar spirit of focal attention [79]. Both MLP architectures and our focal modulation network are attention-free. However, focal modulation with multi-level context aggregation naturally captures the structures in both short- and long-range, and thus achieves much better accuracy-efficiency trade-off.

**Convolutions**. ConvNets have been the primary driver of the renaissance of deep neural networks in computer vision. The field has evolved rapidly since the emerge of VGG [52], InceptionNet [56] and ResNet [27]. Representative works that focus on the efficiency of ConvNets are MobileNet [30], ShuffleNet [90] and EfficientNet [57]. Another line of works aimed at integrating global context to compensate ConvNets such as SE-Net [32], Non-local Network [71], GCNet [2], LR-Net [31] and C3Net [80], *etc*. Introducing dynamic operation is another way to augment ConvNets as demonstrated in Involution [37] and DyConv [10]. Recently, ConvNets strike back from two aspects: ($i$) convolution layers are integrated to SA and bring significant gains [74, 22, 39, 20] or the vice versa [64]; ($ii$) ResNets have closed the gap to ViTs using similar data augmentation and regularization strategies [73], and replacing SA with (dynamic) depth-wise convolution [25, 47] can also slightly surpass Swin. Our focal modulation network also exploits depth-wise convolution as the micro-architecture but goes beyond by introducing a multi-level context aggregation and input-dependent modulation. We will show this new module significantly outperforms raw convolution networks.

## 3 Focal Modulation Network

### 3.1 From Self-Attention to Focal Modulation

Given a visual feature map $\mathbf{X} \in \mathbb{R}^{H \times W \times C}$ as input, a generic encoding process generates for each visual token (query) $\boldsymbol{x}_i \in \mathbb{R}^C$ a feature representation $\boldsymbol{y}_i \in \mathbb{R}^C$ via the interaction $\mathcal{T}$ with its surroundings $\mathbf{X}$ (*e.g.,* neighboring tokens) and aggregation $\mathcal{M}$ over the contexts.

**Self-attention**. The self-attention modules use a late aggregation procedure formulated as

$$\boldsymbol{y}_i = \mathcal{M}_1(\mathcal{T}_1(\boldsymbol{x}_i, \mathbf{X}), \mathbf{X}), \tag{1}$$

where the aggregation $\mathcal{M}_1$ over the contexts $\mathbf{X}$ is performed after the attention scores between query and target are computed via interaction $\mathcal{T}_1$.

**Focal modulation**. In contrast, focal modulation generates refined representation $\boldsymbol{y}_i$ using an early aggregation procedure formulated as

$$\boldsymbol{y}_i = \mathcal{T}_2(\mathcal{M}_2(i, \mathbf{X}), \boldsymbol{x}_i), \tag{2}$$

where the context features are first aggregated using $\mathcal{M}_2$ at each location $i$, then the query interacts with the aggregated feature based on $\mathcal{T}_2$ to form $\boldsymbol{y}_i$.

Comparing Eq. (1) and Eq. (2), we see that ($i$) the context aggregation of focal modulation $\mathcal{M}_2$ amortizes the computation of contexts via a shared operator (*e.g.*, depth-wise convolution), while $\mathcal{M}_1$ in SA is more computationally expensive as it requires summing over non-shareable attention scores for different queries; ($ii$) the interaction $\mathcal{T}_2$ is a lightweight operator between a token and its context, while $\mathcal{T}_1$ involves computing token-to-token attention scores, which has quadratic complexity.

Based on Eq. (2), we instantiate our focal modulation to

$$\boldsymbol{y}_i = q(\boldsymbol{x}_i) \odot m(i, \mathbf{X}), \tag{3}$$

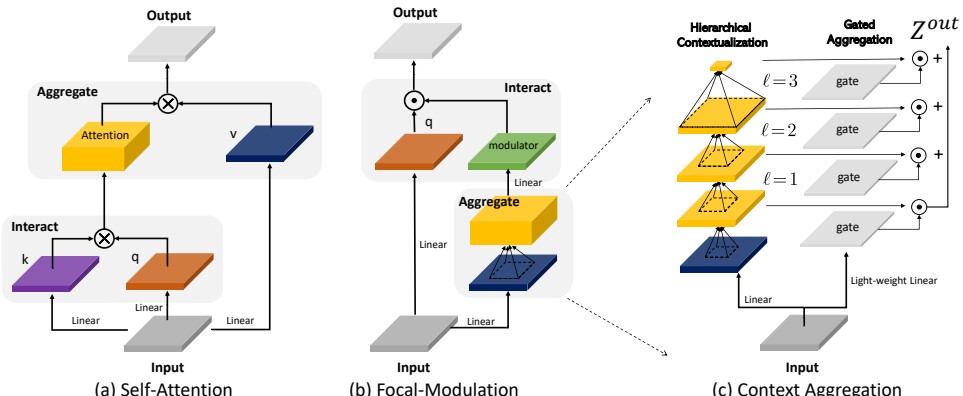

Figure 2: Left: Comparing SA (a) and focal modulation (b) side by side. Right: Detailed illustration of context aggregation in focal modulation (c).

where $q(\cdot)$ is a query projection function and $\odot$ is the element-wise multiplication. $m(\cdot)$ is a context aggregation function, whose output is called *modulator*. Fig. 2(a) and (b) compare self-attention and focal modulation. The proposed focal modulation has the following favorable properties:

- **Translation invariance**. Since $q(\cdot)$ and $m(\cdot)$ are always centered at the query token $i$ and no positional embedding is used, the modulation is invariant to translation of input feature map $\mathbf{X}$.
- **Explicit input-dependency**. The modulator is computed via $m(\cdot)$ by aggregating the local features around target location $i$, hence our focal modulation is explicitly input-dependent.
- **Spatial- and channel-specific**. The target location $i$ as a pointer for $m(\cdot)$ enables spatial-specific modulation. The element-wise multiplication enables channel-specific modulation.
- **Decoupled feature granularity**. $q(\cdot)$ preserve the finest information for individual tokens, while $m(\cdot)$ extracts the coarser context. They are decoupled but combined through modulation.

In what follows, we describe in detail the implementation of $m(\cdot)$ in Eq. (3).

### 3.2 Context Aggregation via $m(\cdot)$

It has been proved that both short- and long-range contexts are important for visual modeling [79, 18, 47]. However, a single aggregation with larger receptive field is not only computationally expensive in time and memory, but also undermines the local fine-grained structures which are particularly useful for dense prediction tasks. Inspired by [79], we propose a multi-scale hierarchical context aggregation. As depicted in Fig. 2 (c), the aggregation procedure consists of two steps: *hierarchical contextualization* to extract contexts from local to global ranges at different levels of granularity and *gated aggregation* to condense all context features at different granularity levels into the modulator.

**Step 1: Hierarchical Contextualization.** Given input feature map $\mathbf{X}$, we first project it into a new feature space with a linear layer $\mathbf{Z}^0 = f_z(\mathbf{X}) \in \mathbb{R}^{H \times W \times C}$. Then, a hierarchical presentation of contexts is obtained using a stack of $L$ depth-wise convolutions. At focal level $\ell \in \{1, ..., L\}$, the output $\mathbf{Z}^\ell$ is derived by:

$$\mathbf{Z}^\ell = f_a^\ell(\mathbf{Z}^{\ell-1}) \triangleq \mathsf{GeLU}(\mathsf{DW\text{-}Conv}(\mathbf{Z}^{\ell-1})) \in \mathbb{R}^{H \times W \times C}, \tag{4}$$

where $f_a^\ell$ is the contextualization function at the $\ell$-th level, implemented via a depth-wise convolution $\mathsf{DW\text{-}Conv}$ with kernel size $k^\ell$ followed by a $\mathsf{GeLU}$ activation function [28]. The use of depth-wise convolution for hierarchical contextualization of Eq. (4) is motivated by its desirable properties. Compared to pooling [83, 32], depth-wise convolution is learnable and structure-aware. In contrast to regular convolution, it is channel-wise and thus computationally much cheaper.

Hierarchical contextualization of Eq. (4) generates $L$ levels of feature maps. At level $\ell$, the effective receptive field is $r^\ell = 1 + \sum_{i=1}^{\ell}(k^\ell - 1)$, which is much larger than the kernel size $k^\ell$. To capture global context of the whole input, which could be high-resolution, we apply a global average pooling on the $L$-th level feature map $\mathbf{Z}^{L+1} = \mathsf{Avg\text{-}Pool}(\mathbf{Z}^L)$. Thus, we obtain in total $(L+1)$ feature maps $\{\mathbf{Z}^\ell\}_{\ell=1}^{L+1}$, which collectively capture short- and long-range contexts at different levels of granularity.

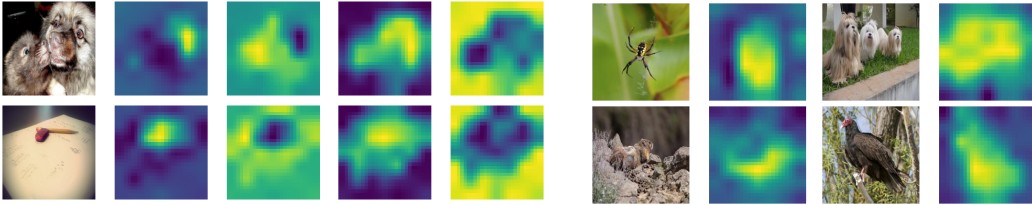

Figure 3: Visualization of gating values $\mathbf{G}$ in Eq. (5) at last layer of our FocalNet ($L = 3$) pretrained on ImageNet-1K. The columns from left to right are input images, gating maps at focal level 1,2,3 and global level.

Figure 4: Visualization of modulator values (corresponding to the right side of $\odot$ in Eq. (6)) at the last layer in FocalNet. The original modulator map is upsampled for display.

**Step 2: Gated Aggregation.**

In this step, the $(L + 1)$ feature maps obtained via hierarchical contextualization are condensed into a modulator. In an image, the relation between a visual token (query) and its surrounding contexts often depends on the content itself. For example, the model might rely on local fine-grained features for encoding the queries of salient visual objects, but mainly global coarse-grained features for the queries of background scenes. Based on this intuition, we use a gating mechanism to control how much to aggregate from different levels for each query. Specifically, we use a linear layer to obtain a spatial- and level-aware gating weights $\mathbf{G} = f_g(\mathbf{X}) \in \mathbb{R}^{H \times W \times (L+1)}$. Then, we perform a weighted sum through an element-wise multiplication to obtain a single feature map $\mathbf{Z}^{out}$ which has the same size as the input $\mathbf{X}$,

$$\mathbf{Z}^{out} = \sum_{\ell=1}^{L+1} \mathbf{G}^\ell \odot \mathbf{Z}^\ell \in \mathbb{R}^{H \times W \times C} \tag{5}$$

where $\mathbf{G}^\ell \in \mathbb{R}^{H \times W \times 1}$ is a slice of $\mathbf{G}$ for the level $\ell$. When visualizing these gating maps in Fig. 3, we surprisingly find our FocalNet indeed learns gathering the context from different focal levels adaptively as we expect. As we can see, for a token on a small object, it focuses more on the fine-grained local structure at low focal level, while a token in a uniform background needs to be aware of much larger contexts from higher levels. Until now, all the aggregation is spatial. To enable the communication across different channels, we use another linear layer $h(.)$ to obtain the modulator map $\mathbf{M} = h(\mathbf{Z}^{out}) \in \mathbb{R}^{H \times W \times C}$. In Fig. 4, we visualize the magnitude of modulator $\mathbf{M}$ at the last layer of our FocalNet. Interestingly, the modulators automatically pay more attention to the objects inducing the category, which implies a simple way of interpreting FocalNets.

**Focal Modulation.** Given the implementation of $m(\cdot)$ as described above, focal modulation of Eq.(3) can be rewritten at the token level as

$$\boldsymbol{y}_i = q(\boldsymbol{x}_i) \odot h\left(\sum_{\ell=1}^{L+1} \boldsymbol{g}_i^\ell \cdot \boldsymbol{z}_i^\ell\right) \tag{6}$$

where $\boldsymbol{g}_i^\ell$ and $\boldsymbol{z}_i^\ell$ are the gating value and visual feature at location $i$ of $\mathbf{G}^\ell$ and $\mathbf{Z}^\ell$, respectively. We summarize the proposed focal modulation in Pytorch-style pseudo code in Algorithm 1. As we can see, it can be easily implemented with a few convolution and linear layers.

### 3.3 Complexity

In focal modulation as Eq. (6), there are mainly three linear projections $q(\cdot)$, $h(\cdot)$, and $f_z(\cdot)$ for $\mathbf{Z}^0$. Besides, it requires a lightweight linear function $f_g(\cdot)$ for gating and $L$ depth-wise convolution $f_a^{\{1,...,L\}}$ for hierarchical contextualization. Therefore, the overall number of learnable parameters is $3C^2 + C(L + 1) + C \sum_\ell (k^\ell)^2$. Since $L$ and $(k^\ell)^2$ are typically much smaller than $C$, the model size is mainly determined by the first term as we will show in Sec. 4. Regarding the time complexity, besides the linear projections and the depth-wise convolution layers, the element-wise multiplications introduce $\mathcal{O}(C(L + 2))$ for each visual token. Hence, the total complexity for a feature map is $\mathcal{O}(HW \times (3C^2 + C(2L + 3) + C \sum_\ell (k^\ell)^2))$. For comparison, a window-wise attention in Swin Transformer with window size $w$ is $\mathcal{O}(HW \times (3C^2 + 2Cw^2))$, where $w$ is the window size.

**Algorithm 1:** Pseudo code for Focal Modulation.

```
# Input/output shape:  (B, H, W, C); Batchsize B; Feature map height H, width W, dim C
# Focal levels:  L; Conv kernel size at level ℓ:  k^ℓ
1 def init( ):
2     pj_in, pj_cxt = Linear(C, 2*C + (L+1)), Conv2d(C, C, 1)
3     hc_layers = [Sequential(Conv2d(C, C, k^ℓ, groups=C), GeLU()) for ℓ in range(L)]
4     pj_out = Sequential(Linear(C, C), Dropout())
5 def forward(x, m=0):
6     x = pj_in(x).permute(0, 3, 1, 2)
7     q, z, gate = split(x, (C, C, L+1), 1)
8     for ℓ in range(L):
9         z = hc_layers[ℓ](z)          # Eq.(4), hierarchical contextualization
10        m = m + z * gate[:, ℓ:ℓ+1]   # Eq.(5), gated aggregation
11    m = m + GeLU(z.mean(dim=(2,3))) * gate[:,L:]
12    x = q * pj_cxt(m)               # Eq.(6), focal modulation
13    return pj_out( x.permute(0, 2, 3, 1) )
```

### 3.4 Network Architectures

We use the same stage layouts and hidden dimensions as in Swin [46] and Focal Transformers [79], but replace the SA modules with the focal modulation modules. We thus construct a series of Focal Modulation Network (FocalNet) variants. In FocalNets, we only need to specify the number of focal levels ($L$) and the kernel size ($k^\ell$) at each level. For simplicity, we gradually increase the kernel size by 2 from lower focal levels to higher ones, *i.e.*, $k^\ell = k^{\ell-1} + 2$. To match the complexities of Swin and Focal Transformers, we design a small receptive field (SRF) and a large receptive field (LRF) version for each of the four layouts by using 2 and 3 focal levels, respectively. We use non-overlapping convolution layers for patch embedding at the beginning (kernel size=$4 \times 4$, stride=4) and between two stages (kernel size=$2 \times 2$, stride=2), respectively.

## 4 Experiment

### 4.1 Image Classification

We compare different methods on ImageNet-1K classification [16]. Following the recipes in [63, 46, 79], we train FocalNet-T, FocalNet-S and FocalNet-B with ImageNet-1K training set and report Top-1 accuracy (%) on the validation set. Training details are described in the appendix.

To verify the effectiveness of FocalNet, we compare it with three groups of methods based on ConvNets, Transformers and MLPs. The results are reported in Table 1. We see that FocalNets outperform the conventional CNNs (*e.g.*, ResNet [27] and the augmented version [73]), MLP architectures such as MLP-Mixer [61] and gMLP [43], and Transformer architectures DeiT [63] and PVT [69]. In particular, we compare FocalNets against Swin and Focal Transformers which use the same architecture to verify FocalNet's stand-alone effectiveness at the bottom part. We see that FocalNets with small receptive fields (SRF) achieve consistently better performance than Swin Transformer but with similar model size, FLOPs and throughput. For example, the tiny FocalNet improves Top-1 accuracy by $0.9\%$ over Swin-Tiny. To compare with Focal Transformers (FocalAtt), we change to large receptive fields (LRF) though it is still much smaller than the one used in FocalAtt. Focal modulation outperforms the strong and sophisticatedly designed focal attention across all model sizes. More importantly, its run-time speed is much higher than FocalAtt by getting rid of many time-consuming operations like rolling and unfolding.

**Model augmentation.** We investigate whether some commonly used techniques for vision transformers can also improve our FocalNets. First, we study the effect of using overlapped patch embedding for downsampling [22]. Following [74], we change the kernel size and stride from $(4, 4)$ to $(7, 4)$ for patch embedding at the beginning, and $(2, 2)$ to $(3, 2)$ for later stages. The comparisons are reported in Table 2. Overlapped patch embedding improves the performance for models of all sizes, with slightly increased computational complexity and time cost. Second, we make our FocalNets deeper but thinner as in [18, 96]. In Table 3, we change the depth layout of our FocalNet-T from 2-2-6-2 to 3-3-16-3, and FocalNet-S/B from 2-2-18-2 to 4-4-28-4. Meanwhile, the hidden dimension at first stage is reduced from 96, 128 to 64, 96, respectively. These changes lead to smaller model sizes and fewer FLOPs, but higher time cost due to the increased number of sequential blocks. It turns out that

| Model | #Params. (M) | FLOPs (G) | Throughput (imgs/s) | Top-1 (%) |
|---|---|---|---|---|
| ResNet-50 [27] | 25.0 | 4.1 | 1294 | 76.2 |
| ResNet-101 [27] | 45.0 | 7.9 | 745 | 77.4 |
| ResNet-152 [27] | 60.0 | 11.0 | 522 | 78.3 |
| ResNet-50-SB [73] | 25.0 | 4.1 | 1294 | 79.8 |
| ResNet-101-SB [73] | 45.0 | 7.9 | 745 | 81.3 |
| ResNet-152-SB [73] | 60.0 | 11.6 | 522 | 81.8 |
| DW-Net-T [25] | 24.2 | 3.8 | 1030 | 81.2 |
| DW-Net-B [25] | 74.3 | 12.9 | 370 | 83.2 |
| Mixer-B/16 [61] | 59.9 | 12.7 | 455 | 76.4 |
| gMLP-S [43] | 19.5 | 4.5 | 785 | 79.6 |
| gMLP-B [43] | 73.4 | 15.8 | 301 | 81.6 |
| ResMLP-S24 [62] | 30.0 | 6.0 | 871 | 79.4 |
| ResMLP-B24 [62] | 129.1 | 23.0 | 61 | 81.0 |
| DeiT-Small/16 [63] | 22.1 | 4.6 | 939 | 79.9 |
| DeiT-Base/16 [63] | 86.6 | 17.5 | 291 | 81.8 |
| PVT-Small [69] | 24.5 | 3.8 | 794 | 79.8 |
| PVT-Medium [69] | 44.2 | 6.7 | 517 | 81.2 |
| PVT-Large [69] | 61.4 | 9.8 | 352 | 81.7 |
| PoolFormer-m36 [83] | 56.2 | 8.8 | 463 | 82.1 |
| PoolFormer-m48 [83] | 73.5 | 11.6 | 347 | 82.5 |
| Swin-Tiny [46] | 28.3 | 4.5 | 760 | 81.2 |
| FocalNet-T (SRF) | 28.4 | 4.4 | 743 | **82.1** |
| Swin-Small [46] | 49.6 | 8.7 | 435 | 83.1 |
| FocalNet-S (SRF) | 49.9 | 8.6 | 434 | **83.4** |
| Swin-Base [46] | 87.8 | 15.4 | 291 | 83.5 |
| FocalNet-B (SRF) | 88.1 | 15.3 | 280 | **83.7** |
| FocalAtt-Tiny [79] | 28.9 | 4.9 | 319 | 82.2 |
| FocalNet-T (LRF) | 28.6 | 4.5 | 696 | **82.3** |
| FocalAtt-Small | 51.1 | 9.4 | 192 | **83.5** |
| FocalNet-S (LRF) | 50.3 | 8.7 | 406 | **83.5** |
| FocalAtt-Base [79] | 89.8 | 16.4 | 138 | 83.8 |
| FocalNet-B (LRF) | 88.7 | 15.4 | 269 | **83.9** |

Table 1: ImageNet-1K classification comparison.

| Model | Overlapped PatchEmbed | #Params. (M) | FLOPs (G) | Throughput (imgs/s) | Top-1 (%) |
|---|---|---|---|---|---|
| FocalNet-T (SRF) | | 28.4 | 4.4 | 743 | 82.1 |
| FocalNet-T (SRF) | ✓ | 30.4 | 4.4 | 730 | **82.4** |
| FocalNet-S (SRF) | | 49.9 | 8.6 | 434 | 83.4 |
| FocalNet-S (SRF) | ✓ | 51.8 | 8.6 | 424 | **83.4** |
| FocalNet-B (SRF) | | 88.1 | 15.3 | 286 | 83.7 |
| FocalNet-B (SRF) | ✓ | 91.6 | 15.3 | 278 | **84.0** |

Table 2: Effect of overlapped patch embedding.

| Model | Depth | Dim. | #Params. | FLOPs | Throughput | Top-1 |
|---|---|---|---|---|---|---|
| FocalNet-T (SRF) | 2-2-6-2 | 96 | 28.4 | 4.4 | 743 | 82.1 |
| FocalNet-T (SRF) | 3-3-16-3 | 64 | 25.1 | 4.0 | 663 | **82.7** |
| FocalNet-S (SRF) | 2-2-18-2 | 96 | 49.9 | 8.6 | 434 | 83.4 |
| FocalNet-S (SRF) | 4-4-28-4 | 64 | 38.2 | 6.4 | 440 | **83.5** |
| FocalNet-B (SRF) | 2-2-18-2 | 128 | 88.1 | 15.3 | 280 | 83.7 |
| FocalNet-B (SRF) | 4-4-28-4 | 96 | 85.1 | 14.3 | 247 | **84.1** |

Table 3: Effect of deeper and thinner networks.

| Model | Img. Size | #Params | FLOPs | Throughput | Top-1 |
|---|---|---|---|---|---|
| ResNet-101x3 [27] | $384^2$ | 388.0 | 204.6 | - | 84.4 |
| ResNet-152x4 [27] | $480^2$ | 937.0 | 840.5 | - | 85.4 |
| ViT-B/16 [19] | $384^2$ | 86.0 | 55.4 | 99 | 84.0 |
| ViT-L/16 [19] | $384^2$ | 307.0 | 190.7 | 30 | 85.2 |
| Swin-Base [46] | $224^2/224^2$ | 88.0 | 15.4 | 291 | 85.2 |
| FocalNet-B | $224^2/224^2$ | 88.1 | 15.3 | 280 | **85.6** |
| Swin-Base [46] | $384^2/384^2$ | 88.0 | 47.1 | 91 | 86.4 |
| FocalNet-B | $224^2/384^2$ | 88.1 | 44.8 | 94 | **86.5** |
| Swin-Large [46] | $224^2/224^2$ | 196.5 | 34.5 | 155 | 86.3 |
| FocalNet-L | $224^2/224^2$ | 197.1 | 34.2 | 144 | **86.5** |
| Swin-Large [46] | $384^2/384^2$ | 196.5 | 104.0 | 49 | 87.3 |
| FocalNet-L | $224^2/384^2$ | 197.1 | 100.6 | 50 | 87.3 |

Table 4: ImageNet-1K finetuning results with models pretrained on ImageNet-22K. Numbers before and after "/" are resolutions used for pretraining and finetuning, respectively.

going deeper improves the performance of FocalNets significantly. These results demonstrate that the commonly used model augmentation techniques developed for vision transformers can be easily adopted to improve the performance of FocalNets.

**ImageNet-22K pretraining.** We investigate the effectiveness of FocalNets when pretrained on ImageNet-22K which contains 14.2M images and 21K categories. Training details are described in the appendix. We report the results in Table 4. Though FocalNet-B/L are both pretrained with $224 \times 224$ resolution and directly transferred to target domain with $384 \times 384$ image size, we can see that they consistently outperform Swin Transformers.

## 4.2 Detection and Segmentation

**Object detection and instance segmentation**. We make comparisons on object detection with COCO 2017 [42]. We choose Mask R-CNN [26] as the detection method and use FocalNet-T/S/B pretrained on ImageNet-1K as the backbones. All models are trained on the 118k training images and evaluated on 5K validation images. We use two standard training recipes, $1\times$ schedule with 12 epochs and $3\times$ schedule with 36 epochs. Following [46], we use the same multi-scale training strategy by randomly resizing the shorter side of an image to $[480, 800]$. Similar to [79], we increase the kernel size $k^\ell$ by 6 for context aggregation at all focal levels to adapt to higher input resolutions. Instead of up-sampling the relative position biases as in [79], FocalNets uses simple zero-padding for the extra kernel parameters. This expanding introduces negligible overhead but helps extract longer range contexts. For training, we use AdamW [48] as the optimizer with initial learning rate $10^{-4}$ and weight decay 0.05. All models are trained with batch size 16. We set the stochastic drop rates to 0.1, 0.2, 0.3 in $1\times$ and 0.3, 0.5, 0.5 in $3\times$ training schedule for FocalNet-T/S/B, respectively.

The results are shown in Table 5. We measure both box and mask mAP, and report the results for both small and large receptive field models. Comparing with Swin Transformer, FocalNets improve the box mAP ($AP^b$) by 2.2, 1.5 and 1.9 in $1\times$ schedule for tiny, small and base models, respectively. In $3\times$ schedule, the improvements are still consistent and significant. Remarkably, the $1\times$ performance of FocalNet-T/B (45.9/48.8) rivals Swin-T/B (46.0/48.5) trained with $3\times$ schedule. When comparing with FocalAtt [79], FocalNets with large receptive fields consistently outperform under all settings

| Backbone | #Params (M) | FLOPs (G) | Mask R-CNN 1x | | | | | | Mask R-CNN 3x | | | | | |
|---|---|---|---|---|---|---|---|---|---|---|---|---|---|---|
| | | | $AP^b$ | $AP_{50}^b$ | $AP_{75}^b$ | $AP^m$ | $AP_{50}^m$ | $AP_{75}^m$ | $AP^b$ | $AP_{50}^b$ | $AP_{75}^b$ | $AP^m$ | $AP_{50}^m$ | $AP_{75}^m$ |
| ResNet50 [27] | 44.2 | 260 | 38.0 | 58.6 | 41.4 | 34.4 | 55.1 | 36.7 | 41.0 | 61.7 | 44.9 | 37.1 | 58.4 | 40.1 |
| PVT-Small[69] | 44.1 | 245 | 40.4 | 62.9 | 43.8 | 37.8 | 60.1 | 40.3 | 43.0 | 65.3 | 46.9 | 39.9 | 62.5 | 42.8 |
| Twins-SVT-S [14] | 44.0 | 228 | 43.4 | 66.0 | 47.3 | 40.3 | 63.2 | 46.8 | 46.8 | 69.2 | 51.2 | 42.6 | 66.3 | 45.8 |
| Swin-Tiny [46] | 47.8 | 264 | 43.7 | 66.6 | 47.7 | 39.8 | 63.3 | 42.7 | 46.0 | 68.1 | 50.3 | 41.6 | 65.1 | 44.9 |
| FocalNet-T (SRF) | 48.6 | 267 | 45.9 (+2.2) | 68.3 | 50.1 | 41.3 | 65.0 | 44.3 | 47.6 (+1.6) | 69.5 | 52.0 | 42.6 | 66.5 | 45.6 |
| FocalAtt-Tiny [79] | 48.8 | 291 | 44.8 | 67.7 | 49.2 | 41.0 | 64.7 | 44.2 | 47.2 | 69.4 | 51.9 | 42.7 | 66.5 | 45.9 |
| FocalNet-T (LRF) | 48.9 | 268 | 46.1 (+1.3) | 68.2 | 50.6 | 41.5 | 65.1 | 44.5 | 48.0 (+0.8) | 69.7 | 53.0 | 42.9 | 66.5 | 46.1 |
| ResNet101 [27] | 63.2 | 336 | 40.4 | 61.1 | 44.2 | 36.4 | 57.7 | 38.8 | 42.8 | 63.2 | 47.1 | 38.5 | 60.1 | 41.3 |
| ResNeXt101-32x4d [77] | 62.8 | 340 | 41.9 | 62.5 | 45.9 | 37.5 | 59.4 | 40.2 | 44.0 | 64.4 | 48.0 | 39.2 | 61.4 | 41.9 |
| PVT-Medium [69] | 63.9 | 302 | 42.0 | 64.4 | 45.6 | 39.0 | 61.6 | 42.1 | 44.2 | 66.0 | 48.2 | 40.5 | 63.1 | 43.5 |
| Twins-SVT-B [14] | 76.3 | 340 | 45.2 | 67.6 | 49.3 | 41.5 | 64.5 | 44.8 | 48.0 | 69.5 | 52.7 | 43.0 | 66.8 | 46.6 |
| Swin-Small [46] | 69.1 | 354 | 46.5 | 68.7 | 51.3 | 42.1 | 65.8 | 45.2 | 48.5 | 70.2 | 53.5 | 43.3 | 67.3 | 46.6 |
| FocalNet-S (SRF) | 70.8 | 356 | 48.0 (+1.5) | 69.9 | 52.7 | 42.7 | 66.7 | 45.7 | 48.9 (+0.4) | 70.1 | 53.7 | 43.6 | 67.1 | 47.1 |
| FocalAtt-Small [79] | 71.2 | 401 | 47.4 | 69.8 | 51.9 | 42.8 | 66.6 | 46.1 | 48.8 | 70.5 | 53.6 | 43.8 | 67.7 | 47.2 |
| FocalNet-S (LRF) | 72.3 | 365 | 48.3 (+0.9) | 70.5 | 53.1 | 43.1 | 67.4 | 46.2 | 49.3 (+0.5) | 70.7 | 54.2 | 43.8 | 67.9 | 47.4 |
| ResNeXt101-64x4d [77] | 102.0 | 493 | 42.8 | 63.8 | 47.3 | 38.4 | 60.6 | 41.3 | 44.4 | 64.9 | 48.8 | 39.7 | 61.9 | 42.6 |
| PVT-Large[69] | 81.0 | 364 | 42.9 | 65.0 | 46.6 | 39.5 | 61.9 | 42.5 | 44.5 | 66.0 | 48.3 | 40.7 | 63.4 | 43.7 |
| Twins-SVT-L [14] | 119.7 | 474 | 45.9 | - | - | 41.6 | - | - | - | - | - | - | - | - |
| Swin-Base [46] | 107.1 | 497 | 46.9 | 69.2 | 51.6 | 42.3 | 66.0 | 45.5 | 48.5 | 69.8 | 53.2 | 43.4 | 66.8 | 46.9 |
| FocalNet-B (SRF) | 109.4 | 496 | 48.8 (+1.9) | 70.7 | 53.5 | 43.3 | 67.5 | 46.5 | 49.6 (+1.1) | 70.6 | 54.1 | 44.1 | 68.0 | 47.2 |
| FocalAtt-Base [79] | 110.0 | 533 | 47.8 | 70.2 | 52.5 | 43.2 | 67.3 | 46.5 | 49.0 | 70.1 | 53.6 | 43.7 | 67.6 | 47.0 |
| FocalNet-B (LRF) | 111.4 | 507 | 49.0 (+1.2) | 70.9 | 53.9 | 43.5 | 67.9 | 46.7 | 49.8 (+0.8) | 70.9 | 54.6 | 44.1 | 68.2 | 47.2 |

Table 5: COCO object detection and instance segmentation results with Mask R-CNN [26].

| Method | Backbone | #Param. | FLOPs | $AP^b$ | $AP_{50}^b$ | $AP_{75}^b$ |
|---|---|---|---|---|---|---|
| C. Mask R-CNN [1] | R-50 [27] | 82.0 | 739 | 46.3 | 64.3 | 50.5 |
| | DW-Net-T [25] | 82.0 | 730 | 49.9 | 68.6 | 54.3 |
| | Swin-T [46] | 85.6 | 742 | 50.5 | 69.3 | 54.9 |
| | FocalNet-T (SRF) | 86.4 | 746 | 51.5 | 70.1 | 55.8 |
| | FocalAtt-T [79] | 86.7 | 770 | 51.5 | 70.6 | 55.9 |
| | FocalNet-T (LRF) | 87.1 | 751 | 51.5 | 70.3 | 56.0 |
| Sparse R-CNN [55] | R-50 [27] | 106.1 | 166 | 44.5 | 63.4 | 48.2 |
| | Swin-T [46] | 109.7 | 172 | 47.9 | 67.3 | 52.3 |
| | FocalNet-T (SRF) | 110.5 | 172 | 49.6 | 69.1 | 54.2 |
| | FocalAtt-T [79] | 110.8 | 196 | 49.0 | 69.1 | 53.2 |
| | FocalNet-T (LRF) | 111.2 | 178 | 49.9 | 69.6 | 54.4 |
| ATSS [88] | R-50 [27] | 32.1 | 205 | 43.5 | 61.9 | 47.0 |
| | Swin-T [46] | 35.7 | 212 | 47.2 | 66.5 | 51.3 |
| | FocalNet-T (SRF) | 36.5 | 215 | 49.2 | 68.1 | 54.2 |
| | FocalAtt-T [79] | 36.8 | 239 | 49.5 | 68.8 | 53.9 |
| | FocalNet-T (LRF) | 37.2 | 220 | 49.6 | 68.7 | 54.5 |

Table 6: A comparison of models with different object detection methods, trained using the $3\times$ schedule.

| Backbone | Crop Size | #Param. | FLOPs | mIoU | +MS |
|---|---|---|---|---|---|
| ResNet-101 [27] | 512 | 86 | 1029 | 44.9 | - |
| Twins-SVT-L [14] | 512 | 133 | - | 48.8 | 50.2 |
| DW-Net-T [25] | 512 | 56 | 928 | 45.5 | - |
| DW-Net-B [25] | 512 | 132 | 924 | 48.3 | - |
| Swin-T [46] | 512 | 60 | 941 | 44.5 | 45.8 |
| FocalNet-T (SRF) | 512 | 61 | 944 | 46.5 | 47.2 |
| FocalAtt-T [79] | 512 | 62 | 998 | 45.8 | 47.0 |
| FocalNet-T (LRF) | 512 | 61 | 949 | 46.8 | 47.8 |
| Swin-S [46] | 512 | 81 | 1038 | 47.6 | 49.5 |
| FocalNet-S (SRF) | 512 | 83 | 1035 | 49.3 | 50.1 |
| FocalAtt-S [79] | 512 | 85 | 1130 | 48.0 | 50.0 |
| FocalNet-S (LRF) | 512 | 84 | 1044 | 49.1 | 50.1 |
| Swin-B [46] | 512 | 121 | 1188 | 48.1 | 49.7 |
| FocalNet-B (SRF) | 512 | 124 | 1180 | 50.2 | 51.1 |
| FocalAtt-B [79] | 512 | 126 | 1354 | 49.0 | 50.5 |
| FocalNet-B (LRF) | 512 | 126 | 1192 | 50.5 | 51.4 |

Table 7: Semantic segmentation on ADE20K [95]. All models are trained with UperNet [75]. MS means multi-scale evaluation.

and cost much less FLOPs. For instance segmentation, we observe the similar trend as that of object detection for FocalNets. To further verify the generality of FocalNets, we train three detection models, Cascade Mask R-CNN [1], Sparse RCNN [55] and ATSS [88] with FocalNet-T as the backbone. We train all models with $3\times$ schedule, and report the box mAPs in Table 6. As we can see, FocalNets bring clear gains to all three detection methods over the previous SoTA methods.

**Semantic segmentation**. We benchmark FocalNets on semantic segmentation, a dense prediction task that requires fine-grained understanding and long-range interactions. We use ADE20K [95] for our experiments and follow [46] to use UperNet [75] as the segmentation method. With FocalNet-T/S/B trained on ImageNet-1K as the backbones, we train UperNet for 160k iterations with input resolution $512 \times 512$ and batch size 16. For comparisons, we report both single- and multi-scale (MS) mIoU. Table 7 shows the results with different backbones. FocalNet outperforms Swin and Focal Transformer significantly under all settings. Even for the base models, FocalNet (SRF) exceeds Swin Transformer by $2.1$ and $1.4$ at single- and multi-scale, respectively. Compared with Focal Transformer, FocalNets outperform Focal Transformer, with a larger gain than that of Swin Transformer, and consume much less FLOPs. These results demonstrate the superiority of FocalNets on the pixel-level dense prediction tasks, in addition to the instance-level object detection task.

Given the superior results for FocalNets on segmentation tasks shown in Table 7, we further investigate its effectiveness while scaling up. Particularly, to fairly compare with Swin-L pretrained on ImageNet-22K with $384\times384$, we also pretrain our FocalNet-L on ImageNet-22K with $384\times384$ with 3 focal levels and kernel sizes $[3, 5, 7]$. We use Mask2former [12] for semantic segmentation on ADE20K and panoptic segmentation on COCO. As shown in Table 8, FocalNet-L achieves superior performance to

| Backbone | Method | #Param | mIoU | +MS |
|---|---|---|---|---|
| HRNet-w48 [54] | OCRNet [85] | 71M | 45.7 | - |
| ResNeSt-200 [86] | DLab.v3+ [6] | 88M | 48.4 | - |
| Swin-B [46] | UperNet [75] | 121M | 48.1 | 49.7 |
| Twins-SVT-L [14] | UperNet [75] | 133M | 48.8 | 50.2 |
| MiT-B5 [76] | SegFormer [76] | 85M | 51.0 | 51.8 |
| ViT-L/16† [19] | SETR [94] | 308M | 50.3 | - |
| Swin-L† [46] | UperNet [75] | 234M | 52.1 | 53.5 |
| ViT-L/16† [19] | Segmenter [53] | 334M | 51.8 | 53.6 |
| Swin-L† [46] | K-Net [89] | - | - | 54.3 |
| Swin-L† [46] | PatchDiverse [21] | 234M | 53.1 | 54.4 |
| VOLO-D5 [84] | UperNet [75] | - | - | 54.3 |
| Focal-L† | UperNet [75] | 240M | 54.0 | 55.4 |
| CSwin-L† | UperNet [75] | 208M | 54.0 | 55.7 |
| BEIT-L† | UperNet [75] | 441M | 56.7 | 57.0 |
| Swinv2-G‡ [45] | UperNet [75] | >3.0B | 59.1 | - |
| ViT-Adapter-L† [11] | Mask2Former [12] | 568M | 58.3 | 59.0 |
| Swin-L† | Mask2Former [12] | 216M | 56.4 | 57.7 |
| Swin-L-FaPN† | Mask2Former [12] | - | 56.1 | 57.3 |
| Swin-L-SeMask† [33] | Mask2Former [12] | - | 57.0 | 58.2 |
| FocalNet-L† (Ours) | Mask2Former [12] | 218M | 57.3 | 58.5 |

Table 8: Systematic comparisons of semantic segmentation on ADE20K validation set. † indicates pretraining with ImageNet-22K and ‡ means using extra data additionally.

| Backbone | Method | #Param. | PQ | AP | mIoU |
|---|---|---|---|---|---|
| ResNet-50 [27] | DETR [3] | - | 43.4 | - | - |
| ResNet-50 [27] | K-Net [89] | - | 47.1 | - | - |
| ResNet-50 [27] | Panoptic SegFormer [40] | 47M | 50.0 | - | - |
| ResNet-50 [27] | Mask2Former [12] | 44M | 51.9 | 41.7 | 62.4 |
| PVTv2-B5 [70] | Panoptic SegFormer [40] | 101M | 54.1 | - | - |
| Swin-T [46] | MaskFormer [13] | 42M | 47.7 | 33.6 | 60.4 |
| Swin-B [46] | MaskFormer [13] | 102M | 51.1 | 37.8 | 62.6 |
| Swin-T [46] | Mask2Former [12] | 47M | 53.2 | 43.3 | 63.2 |
| Swin-B [46] | Mask2Former [12] | 107M | 55.1 | 45.2 | 65.1 |
| Swin-L† [46] | MaskFormer [13] | 212M | 52.7 | 40.1 | 64.8 |
| Swin-L† [46] | Panoptic SegFormer [40] | - | 55.8 | - | - |
| Swin-L† [46] | Mask2Former [13] (200 queries) | 216M | 57.8 | **48.6** | **67.4** |
| Focal-L† (Ours) | Mask2Former [13] (200 queries) | 226M | **57.9** | 48.4 | 67.3 |

Table 9: Panoptic segmentation on COCO [42]. † means pretraining with ImageNet-22K. All models evaluated on minival with single-scale. PQ, AP and mIoU are three metrics for measuring the panoptic segmentation, instance segmentation and semantic segmentation, respectively.

| Model | Formula | #Param. | FLOPs | Throughput | Top-1 |
|---|---|---|---|---|---|
| **FocalNet-T (LRF)** | $\boldsymbol{y}_i = q(\boldsymbol{x}_i) \odot h(\sum_{\ell=1}^{L+1} \boldsymbol{g}_i^\ell \cdot \boldsymbol{z}_i^\ell)$ | 28.6 | 4.49 | 696 | 82.3 |
| → **Depth-width ConvNet** | $\boldsymbol{y}_i = q(\mathsf{GeLU}(h(\boldsymbol{z}_i^L)))$ | 28.6 | 4.47 | 738 | 81.6 (-0.7) |
| → **Pooling Aggregator** | $\boldsymbol{y}_i = q(\boldsymbol{x}_i) \odot h(\sum_{\ell=1}^{L+1} \boldsymbol{g}_i^\ell \cdot \mathsf{Avg\text{-}Pool}(\boldsymbol{z}_i^{\ell-1}))$ | 28.3 | 4.37 | 676 | 80.5 (-1.8) |
| → **Global Pooling Aggregator** | $\boldsymbol{y}_i = q(\boldsymbol{x}_i) \odot h(\boldsymbol{g}_i \cdot \mathsf{Avg\text{-}Pool}(f_z(X)))$ | 28.3 | 4.36 | 883 | 75.7 (-6.7) |
| → **Multi-scale Self-Attention (QKV first)** | $\boldsymbol{y}_i = MHSA(\boldsymbol{x}_i, \boldsymbol{z}_i^1, ..., \boldsymbol{z}_i^{L+1}), f_z, q, h = Identity(\cdot)$ | 28.6 | 4.61 | 456 | 81.5 (-0.8) |
| → **Multi-scale Self-Attention (QKV later)** | $\boldsymbol{y}_i = MHSA(\boldsymbol{x}_i, \boldsymbol{z}_i^1, ..., \boldsymbol{z}_i^{L+1}), f_z, q, h = Identity(\cdot)$ | 28.6 | 7.26 | 448 | 80.8 (-1.5) |
| → **Sliding-window Self-Attention** | $\boldsymbol{y}_i = MHSA(\boldsymbol{x}_i, \mathcal{N}(\boldsymbol{x}_i)), |\mathcal{N}(\boldsymbol{x}_i)| = 7 \times 7 - 1$ | 28.3 | 4.49 | 103 | 81.5 (-0.8) |

Table 10: Performance for different FocalNet model variants.

Swin-L with similar model size and same pretraining data. We note that the methods in gray font like Swinv2-G and ViT-Adapter-L achieve better performance but use much more parameters and training data. We leave the further scaling-up of our FocalNets as future work. In Table 9, we compare different models for panoptic segmentation on COCO with 133 categories. Our FocalNet-L slightly outperforms Swin-L on PQ, rendering a new state-of-the-art for panoptic segmentation. These results clearly demonstrate the effectiveness of our FocalNets for various segmentation tasks.

## 4.3 Network Inspection

**Model Variants**. We compare in Table 10 six different model variants derived from FocalNet.

- **Depth-wise ConvNet.** It feeds the feature vectors at the top level $L$ to a two-layer MLP. The resultant model is close to DW-Net [25]. Although it can achieve 81.6%, surpassing Swin (81.3%), it underperforms FocalNet by 0.7%. FocalNet uses depth-wise convolutions as a component but differently for aggregating contexts, which is then used to modulate each individual tokens.

- **Pooling Aggregator.** It replaces the depth-wise convolution module with average pooling, and is similar to MetaFormer [83] in terms of token aggregation. Average pooling has slightly lower complexity but leads to a significant drop of accuracy by 1.8%. Compared with depth-wise convolution, pooling is permutation-invariant and thus incapable of capturing visual structures.

- **Global Pooling Aggregator.** It removes local aggregations at all levels and only keeps the global one ($\mathbf{Z}^{L+1}$). This variant resembles SENet [32]. It turns out that global context alone is insufficient for visual modeling, leading to a significant 6.7% drop.

- **Multi-scale Self-Attention.** Given the summarized tokens at different levels, a straightforward way to combine them is performing a SA among all of them. We have developed two SA methods: computing $q, k, v$ before and after aggregation, respectively. Both methods result in visible performance drop and increase the run time latency, compared to FocalNet.

- **Sliding-window Self-Attention.** Finally, we apply a sliding-window SA for each visual token within a window. Since it involves dense interactions for each fine-grained tokens, the time and memory cost explodes, and the performance is worse than FocalNet.

| Model | FLOPs | Throughput | Top-1 | $AP^b$ | $AP^m$ |
|---|---|---|---|---|---|
| FocalNet-T (LRF) | 4.48 | 696 | 82.3 | 46.2 | 41.6 |
| Additive | 4.49 | 670 | 81.5 (-0.8) | 45.6 (-0.6) | 41.1 (-0.5) |
| No global pool | 4.48 | 683 | 82.0 (-0.3) | 45.8 (-0.4) | 41.2 (-0.4) |
| Top-only | 4.49 | 698 | 81.9 (-0.4) | 45.7 (-0.5) | 41.2 (-0.4) |
| No gating | 4.48 | 707 | 81.9 (-0.4) | 45.6 (-0.6) | 41.1 (-0.5) |

Table 11: Component analysis for focal modulation. Four separate changes are made to the original FocalNet. Throughput is reported on image classification. All variants have almost the same size (28.6M) as the default model.

| Levels (Kernels) | Receptive Field | #Param. | FLOPs | Throughput | Top-1 |
|---|---|---|---|---|---|
| 2 (3-5) | 7 | 28.4 | 4.41 | 743 | 82.1 |
| 3 (3-5-7) | 13 | 28.6 | 4.49 | 696 | 82.3 |
| 0 (n/a) | 0 | 28.3 | 4.35 | 883 | 75.7 |
| 1 (3) | 3 | 28.3 | 4.37 | 815 | 82.0 |
| 4 (3-5-7-9) | 21 | 29.0 | 4.59 | 592 | 82.2 |
| 1 (13) | 13 | 28.8 | 4.59 | 661 | 81.9 |

Table 12: Model performance with number of focal levels $L$. "Receptive Field" refers to effective receptive field at the top level regardless of the global average pooling.

**Component Analysis**. Here we ablate FocalNet to study the relative contribution of each component. The result is reported in Table 11, where we investigate the impact of the following model architecture changes on model performance:

- **Replacing Multiplication with Addition**: we change the element-wise multiplication to addition in Eq. (6), which converts the modulator into a bias term. This leads to 0.7% accuracy drop, which indicates that element-wise multiplication is a more powerful way of modulation than addition.

- **No Global Aggregation**: we remove the top global average pooling in focal modulation. It hurts the performance by 0.3%. Even though the hierarchical aggregation already covers a relatively large receptive field, global information ($\mathbf{Z}^{L+1}$) is still useful for capturing global context.

- **Top-only Aggregation**: Instead of aggregating the feature maps from all focal levels, we only use the top level map. In this case, the features at lower levels that are more "local" and "fine-grained" are completely discarded. This change leads to 0.4% performance drop, which verifies our hypothesis that features at different levels and spatial scopes compensate each other.

- **None-gating Aggregation**: We remove the gating mechanism when aggregating the multiple levels of feature maps. This causes 0.4% drop. As we discussed earlier, the dependencies between visual token (query) and its surroundings differ based on the query content. The proposed gating mechanism helps the model to *adaptively* learn where and how much to gather.

In Table 12, we study the effect of varying the focal level (*i.e.* the number of depth-wise convolution layers $L$). In our experiments reported above, the results show that large receptive field in general achieves better performance (LRF *v.s.* SRF). Here, we investigate by further altering $L$. In additional to setting $L = 2$ and 3, we also try $L = 0$, $L = 1$, and $L = 4$. Accordingly, increasing $L$ brings slight improvement and finally reaches a plateau. Surprisingly, a single level with kernel size 3 can already obtain a decent performance. When we increase the single-level kernel size from 3 to 13, there is a slight 0.1% drop, and a 0.4% gap to the one with three levels but same size of receptive field (second row). This indicates that simply increasing the receptive field does not necessarily improve the performance, and a hierarchical aggregation for both fine- and coarse-grained contexts is crucial.

At last, we study whether our focal modulation can fit the monolithic architectures like ViTs. We replace all SA modules in ViTs with focal modulation to construct monolithic FocalNet-T/S/B. We use patch size 16 and three focal levels with kernel sizes 3,5 and 7, so that the effective receptive field is close to the global SA in ViT. As shown in Table 13, FocalNets consistently outperform ViTs, with comparable FLOPs and inference speed.

| Model | Dim | #Param. | FLOPs | Th. (imgs/s) | Top-1 |
|---|---|---|---|---|---|
| ViT-T/16 | 192 | 5.7 | 1.3 | 2834 | 72.2 |
| FocalNet-T/16 | 192 | 5.9 | 1.1 | 2334 | **74.1** (+1.9) |
| ViT-S/16 | 384 | 22.1 | 4.6 | 1060 | 79.9 |
| FocalNet-S/16 | 384 | 22.4 | 4.3 | 920 | **80.9** (+1.0) |
| ViT-B/16 | 768 | 86.6 | 17.6 | 330 | 81.8 |
| FocalNet-B/16 | 768 | 87.2 | 16.9 | 300 | **82.4** (+0.6) |

Table 13: Comparisons between FocalNet and ViT both with monolithic architectures.

## 5 Conclusion

We have proposed *focal modulation*, a new mechanism that enables input-dependent token interactions for visual modeling. It consists of a hierarchical contextualization to gather for each query token its contexts from short- to long-ranges, a gated aggregation to adaptively gather contexts based on the query content, followed by a simple modulation. With *focal modulation*, we built a series of simple attention-free Focal Modulation Networks (FocalNets). Extensive experiments show that FocalNets significantly outperform the SoTA SA counterparts (*e.g.,* Swin and Focal Transformer) with similar time-/memory-cost on the tasks of image classification, object detection and semantic segmentation.

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
