# OpenReview forum: "Focal Modulation Networks"
_NeurIPS.cc/2022/Conference — NeurIPS 2022 Accept_

### Official Review · Reviewer_1dzg · 2022-07-09

**Rating:** 6
**Confidence:** 4
**Soundness:** 4 excellent
**Presentation:** 3 good
**Contribution:** 3 good

**Summary:**

This paper proposes a focal modulation module to replace the attention module in Transformers. Specifically, this module contains Hierarchical Contextualization (several layers of depthwise convolutions), Gated Aggregation, and element-wise modulation to fuse information from the token itself and the context. Actually, the element-wise modulation can also be regarded as a gate mechanism. Thus, this module sounds like a convolution + gate mechanism. This paper does extensive experiments on various vision tasks to show the advantages of the proposed modules.

**Questions:**

See weaknesses.

**Limitations:**

Yes

**Strengths And Weaknesses:**

Strengths:

1) The writing of this paper is clear and integrated.

2) The experiments in this paper are extensive, including several vision tasks and ablation studies. It is convinced that Focal Net is better than Swin and Focal Transformers.

Neutrality: Focal Modulation can be regarded as convolutions plus gate mechanisms. The idea is OK, not incremental but not good enough. Thus, I write the idea aspect into neutrality.

Weaknesses:

In ablation table 9, about the fusion between token itself and context, it only shows the experiment that replaces Multiplication with Addition. What about totally removing the query branch and moving the parameters and computation to other components?

---

> ### Author Response · Authors · 2022-08-02
> **Response to reviewer 1dzg**
>
> We thank the reviewer for recognizing the soundness of our work! We also thank the reviewer for all the valuable comments and answer the questions one by one below.
>
> **Q1. Performance when removing query branch and moving parameters to other components**
>
> Thanks for the suggestion! In Table 8 second row, we actually had a model variant that removes the query branch and uses the query embedding layer $q$ to encode the output of the $L=3$ stacked depth-wise convolution layers formulated as:
>
> $y_i = q(GeLU(h(Z^L_i)))$
>
> This variant uses exactly the same amount of parameters as our FocalNet (28.6M). Accordingly, it achieves 81.6% top-1 accuracy, with 0.7% gap to our FocalNet. This indicates that the mechanism matters a lot even with the same amount of data. Together with the addition-based model, these experiments clearly demonstrate the effectiveness of our proposed focal modulation mechanism for encoding visual inputs.
>
> **Q2. FocalNet can be regarded as convolutions plus gating mechanism.**
>
> Thanks for sharing this perspective! We agree with the reviewer that our proposed focal modulation can be regarded as convolutions plus gate mechanisms at a high level. However, our proposed focal modulation is unique in that each component in it is clearly motivated to attain the favorable properties summarized in lines 124-131. We proposed hierarchical contextualization to efficiently and effectively capture the short- and long-range context surrounding each token. We used gated aggregation to adaptive gather the contexts at different granularities depending on the input. We further deployed a modulation mechanism to modulate individual tokens at the finest grain in spatial- and channel-specific manner. According to the experimental results, our FocalNets outperform previous methods consistently across different settings and tasks. Based on a comprehensive study, we believe our FocalNets could stand out as a new component for building a generic vision backbone.

---

### Official Review · Reviewer_pZvC · 2022-07-11

**Rating:** 6
**Confidence:** 3
**Soundness:** 4 excellent
**Presentation:** 4 excellent
**Contribution:** 3 good

**Summary:**

The paper propose a focal modulation module that is more effective and efficient for modeling
token interactions. Their main contributions are proposing an efficient way to input-dependent long-range contextual interactions. The authors conduct experiments on tasks of image classification, detection, and segmentation. The experiments' results show the SoTA performance.

**Questions:**

### questions
(1) About training consuming. Will Context Aggregation cause low latency?
(2) About Table 10. It seems that low receptive field impacts the performance little. What if bottom-only with small kernels? Is it means the context is not important? It's confusing.
(3) About Network Inspection. A Global Pooling Aggregator will lead to a significant drop. May because of low parameters? If we replace local aggregations with several simple downsample convolutions, does it will work?


**Limitations:**

yes

**Strengths And Weaknesses:**

+ The performance achieves SOTA on almost tasks.
+ The paper is well-written.
+ The authors conduct abundant ablation studies to validate the effectiveness of each design in the proposed method.

---

> ### Author Response · Authors · 2022-08-02
> **Response to reviewer pZvC**
>
> We thank the reviewer for acknowledging the soundness and contribution of our work! We also thank the reviewer for all the valuable comments and answer the questions one by one below.
>
> **Q1. Training cost and latency for context aggregation.**
>
> Our context aggregation consists of hierarchical contextualization and gated aggregation as depicted in Fig. 2c, which are implemented by a stack of depth-wise convolution followed by element-wise multiplication. Both operations are implemented with built-in pytorch functions which have been highly optimized as we can see from the code in our supplementary material. Moreover, since FocalNet exploits a hierarchical stack of depth-wise convolutions for contextualization, it uses relatively smaller kernels but still capture long-range visual interactions. As such, our FocalNet has a very friendly training cost. For example, with 32 Nvidia V100 GPUs, training FocalNet-Tiny on ImageNet-1K for 300 epochs costs less than one day for both SRF and LRF settings. Regarding object detection, training Mask R-CNN for 12 epochs (1x) cost less than 10 hours with 16 Nvidia V100 GPUs for both FocalNet-T (SRF) and FocalNet-T (LRF). We will cover this information in our revision. In terms of inference latency, we reported the throughputs for our FocalNets on image classification in Tables 1, 2, 3, and 4. Accordingly, it has comparable run-time speed as Swin but better performance. Meanwhile, it outperforms Focal Transformer while running much faster.
>
> **Q2. Importance of context in focal modulation.**
>
> In Table 10, we conducted experiments with FocalNet-T by using different numbers of focal levels, from 0 to 4. Accordingly, setting #focal level to 0 causes a significant drop of the top-1 accuracy on imagenet-1k. This indicates that contextualization is crucial to the final performance in that it enables the interactions between individual visual tokens with their surrounding context. When using more than 0 focal levels, individual visual tokens are modulated with the contexts at different ranges and the performance is improved to 82.0, 82.1, 82.3 and 82.2 for #focal levels=1,2,3 and 4, respectively. The gaps seem small but the improvements are non-trivial when using the 300 epoch training regime with strong data augmentations, as observed in many previous works. Additionally, according to the experiments on object detection and segmentation in Tables 5,6 and 7, FocalNets with three focal levels (LRF) in general outperform FocalNets with two focal levels (SRF). To further study this aspect, we applied our FocalNet-Tiny with a single focal level (3x3 kernel) to object detection and observed a significant drop of box mAP from 46.2 to 45.3 when trained with Mask R-CNN 1x. All these experimental results indicate performing modulation with the surrounding context is important and a single local context is not sufficient to get good performance across different tasks. We recommend using two or three focal levels as a generic setting for a variety of vision tasks.
>
> **Q3. Global pooling and downsample convolutions for aggregation.**
>
> As we can see from the #Param. column in Table 8, the model with a global pooling aggregator consumes a similar amount of parameters as the default FocalNet (28.3M v.s. 28.6M), and the small amount of 0.3M parameters are introduced by the stacked depth-wise convolution layers for contextualization and gating. With a similar amount of parameters, the comparison clearly indicates that extracting the modulator specifically for each visual token based on the surrounding context is critical to the final good performance. A simple global pooling aggregator totally discards the spatial information at each location and thus significantly weakens the representation ability. As suggested by the reviewer, we tried another variant of our FocalNet. Specifically, instead of using depth-wise conv with stride=1 for local aggregation, we change the stride to 2 at all focal levels to downsample the input feature map progressively, followed by a global average pooling on the downsampled feature map. This variant uses the same amount of parameters as our FocalNet. According to the experiment with tiny model, this variant reaches 76.8% top-1 accuracy, which is slightly better than the Global Pooling Aggregator (75.7%), but still significantly lags behind our default FocalNet (82.3%). As a result, we recommend using a spatial- and channel-specific modulation as proposed in our FocalNet.

---

> > ### Comment · Reviewer_pZvC · 2022-08-09
> > **Thank you for your responses.**
> >
> > Thank you for your responses. They addressed my concerns.

---

### Official Review · Reviewer_RTBL · 2022-07-11

**Rating:** 7
**Confidence:** 4
**Soundness:** 3 good
**Presentation:** 3 good
**Contribution:** 3 good

**Summary:**

This paper proposes Focal Modulation, which uses context at multiple spatial scales from a stack of convolutions combined with gated aggregation to produce a modulation.  This modulates each query point through elementwise multiplication.  Focal Modulation is evaluate as a drop-in replacement for Self-Attention.  The method is tested in several experiments against strong baselines, and has a thorough ablation analysis.

## Update after author response

Thank you for the responses and further experiments.  To be clear, I was mostly concerned with the kernel size experiment as a way to glean the variance of other results.  Since you have provided direct experiments to address that concern, I am more confident that this is a paper worthy of acceptance.

**Questions:**

I would like to see some notion of noise in the results based on random seed.  It is tempting to draw conclusions from the results as presented, but some of the effect sizes are small.

**Limitations:**

The limitations and societal impact are well addressed.

**Strengths And Weaknesses:**

The paper proposes a novel method for improving computer vision architectures based on visual transformers.  Focal Modulation is a clever way to include expanded context at each layer while simultaneously removing expensive pairwise interaction terms present in self-attention.  The method is described clearly, the diagrams are illustrative, and the experimental evaluations and ablations are thorough.  This paper is a solid contribution to an important field of computer vision. However, I would like to see some better estimate of variance of results.  For example, in Figure 5 in the appendix, the mAP results vary considerably depending on the setting of the kernel size.  There seems to be little relationship between the kernel size and the mAP, but the mAP varies considerably (41.2 to 41.6).

---

> ### Author Response · Authors · 2022-08-02
> **Response to reviewer RTBL**
>
> We thank the reviewer for recognizing the novelty and strengths of our work! We also thank the reviewer for all the valuable comments and answer the questions one by one below.
>
> **Q1. Object detection performance with different kernel sizes.**
>
> Thanks for pointing it out! The main goal of the experiment in Fig.5 is to investigate whether our FocalNets can use larger kernel sizes for object detection even though the model was trained on imagenet-1k with small kernel size. As we described in lines 638-648, we varied the kernel sizes at the first focal level for our FocalNet-T (LRF) from default 3 to 5,7,9,11,13,15 using a simple zero-padding and the kernel sizes at the following levels are also increased by the same amount. Accordingly, a kernel size of 5,7 or 9 at the first focal level is better than a kernel size of 3 by default. Since the resolution for dense prediction tasks such as object detection and image segmentation is typically higher than that in image classification, a relatively larger kernel size can help to capture more long-range context. However, keeping increasing the kernel sizes will not further bring gain since it will also sacrifice the fine-grained local structures in local regions. To summarize, these experiments demonstrate that: 1) we can flexibly change the focal kernel sizes for our FocalNets to adapt to different downstream tasks **without** any re-pretraining on image classification; 2) using slightly larger kernel sizes can bring gains for object detection and instance segmentation considering the increased image resolution, but keeping increasing it will also make the model overlook the local fine-grained context. As such we recommend selecting one from [3,5,7,9] as the kernel size at the first focal level for our FocalNet when finetuning it on downstream tasks.
>
> **Q2. Some notion of noise in the results based on random seeds.**
>
> Thanks for this suggestion! As suggested, we study this aspect based on two experiments. For image classification, we trained the FocalNet-T (SRF) and FocalNet-T (LRF) with three different random seeds (0,1,2), respectively. We found the variance is pretty small. Specifically, we got 82.1, 82.1 and 82.2 for FocalNet-T (SRF) and 82.3, 82.2, and 82.3 for FocalNet-T (LRF). When adapting the models to object detection with Mask R-CNN 1x schedule using different random seems but the same kernel sizes. We observed around 0.1 stddev on box mAP. Note that these variances do not break any claims about our FocalNets compared with previous works and also demonstrate the robustness of our FocalNets to different initializations. Due to limited time, we were not able to perform a thorough analysis on all model sizes and tasks during this rebuttal, but we will present a more comprehensive analysis in our revision.

---

### Official Review · Reviewer_EqAk · 2022-07-12

**Rating:** 6
**Confidence:** 4
**Soundness:** 3 good
**Presentation:** 4 excellent
**Contribution:** 3 good

**Summary:**

In this paper, the authors introduce a novel Transformer (or Network) architecture, termed as Focal Modulation Network. FocalNet deals with the problem of efficient long-range feature modeling. Different from window-wise self-attention (Swin) and focal attention, FocalNet adaptively aggregates surrounding tokens from different levels of granularity. This is novel and interesting. Extensive experiments on MS COCO, ImageNet, and ADE20k demonstrate the effectiveness of the proposed method. In summary, the idea is interesting, and the performance is promising; I would like to recommend acceptance.

**Questions:**

See above weaknesses.

Also, a question is why the authors consider the contextualizations hierarchically rather than parallelly? I thought the contextualizations would be independent when I read the introduction for the first time.

**Limitations:**

The authors adequately addressed the limitations and potential negative societal impact of their work.

**Strengths And Weaknesses:**

**Strengths:**
1. I enjoy reading this paper, the writing, and the presentation.
2. The idea is interesting and novel. Considering different levels of granularity makes sense in image processing.
3. The experimental results are promising, demonstrating state-of-the-art performance.

**Weaknesses:**
1. From L124-L132, it is easy to understand the Translation invariance and Decoupled feature granularity, but why input-dependency is considered as an advantage? Also,  Spatial- and channel-specific is achieved by depth-wise operations and cannot be considered as an advantage.
2. I like the idea of different levels of granularity, but this can simply be considered as a multi-scale depth-wise convolution in the implementation (Fig. 2c), which limits the novelty. Multi-scale/ multi-branch always works and introduces no novelty.
3. Some important papers are missing (not compared), like ConvNeXt, that only uses depth-wise convolution as well, and MixFormer, which can be considered as a strong baseline. Both are published in CVPR'22 and released much earlier than the submission dealine. The authors should cite and compare these baselines.
[1] Liu, Zhuang, et al. "A convnet for the 2020s." Proceedings of the IEEE/CVF Conference on Computer Vision and Pattern Recognition. 2022.
[2] Chen, Qiang, et al. "MixFormer: Mixing Features across Windows and Dimensions." Proceedings of the IEEE/CVF Conference on Computer Vision and Pattern Recognition. 2022.

---

> ### Author Response · Authors · 2022-08-02
> **Response to reviewer EqAk**
>
> We thank the reviewer for acknowledging the novelty and contributions of our work! We also thank the reviewer for all the valuable comments and answer the questions one by one below.
>
> **Q1. Claimed favorable properties of focal modulation**
>
> First of all, we agree with the reviewer that the listed four properties in lines 124-132 are not unique advantages of focal modulation compared with previous methods, such as self-attention or depth-wise conv. Therefore we carefully stated them as favorable properties instead of advantages, and our proposed focal modulation is unique in that it simultaneously possesses all these favorable properties.
>
> We think the input-independent computation to be a favorable property because the model can adaptively compute the feed-forwarding parameters depending on the local inputs. And this property is arguably one of the advantages of self-attention compared with convolution with a shared kernel [19]. This advantage is demonstrated by our experiments in Table 8, solely using depth-wise convolution and a linear projection (second row) without the input-dependent modulation degrades the top-1 accuracy by 0.7%. It is further demonstrated in our visualizations of the learned modulator in Fig. 4 and the supplementary. Our model computes the modulator adaptively and automatically pays more effort to modulate object regions for better visual recognition.
>
> We think the spatial- and channel-specific modulation is another favorable property because it enables the finest interactions at different spatial locations and feature channels. In our network inspection experiments in Table 8, we replaced the spatial-specific modulator with a single globally pooled feature (fourth row) and found the top-1 accuracy dropped significantly by 6.7%. Moreover, inspired by the reviewer, we added another experiment that applied the averaged modulator to all feature channels and found the top-1 accuracy dropped from 82.3% to 80.8%. Clearly, compared with its counterparts, spatial- and channel-specific modulation enables finest-grained interactions and leads to significantly better performance.
>
> **Q2. Relation to multi-scale depth-wise convolution**
>
> We fully agree with the reviewer that the depth-wise convolution itself does not introduce novelty. As we discussed in lines 100-103, our proposed focal modulation mechanism uses depth-wise conv as the micro-component, but goes beyond it in three aspects:
>
> * **Component is different**. We used depth-wise conv but further developed hierarchical contextualization, gated aggregation and input-dependent modulation for each visual query. As the title entails, the key innovation of our submission lies on the focal modulation module as a whole (Fig. 2b) to enable efficient and effective interactions between different granularities of contexts and individual tokens in comparison with the dense interactions in self-attention. The reason why we used depth-wise conv is that it can capture the local structure well yet cost fewer computations.
>
> * **Mechanism is different**. Previously the depth-wise conv is used to process the input features and generate the output features in a feed-forward manner, such as CvT [59], DWNet [19] and ConvNext ([47] in the supplementary document). In contrast, in focal modulation, the multi-scale depth-wise conv is dedicated to extracting the short- and long-range contexts surrounding individual tokens which are then processed as a modulator to modulate the query token at each location. As such, the role of depth-wise conv in previous work and our focal modulation are different. The visualization results in Fig. 3 and 4 do show that focal modulation extracts the information at different granularities and composes the modulator adaptively given different input contents.
>
> * **Performance is better**. In our experiments, we did a thorough comparison between our proposed focal modulation module with the variants which solely rely on depth-wise convolutions. In Table 8 second row, when we replaced our focal modulation with a stack of depth-wise conv layers and linear projection layer, the imagenet top-1 accuracy drops by 0.7%. In Table 9 second row, if we replace the modulation with simple addition, the performance drops by 0.8% on imagenet. Note that the parameters used in these variants are the same. These results clearly demonstrated that our focal modulation is more powerful than multi-scale depth-wise convs.
>
> Based on the above analysis, we would like to note that our focal modulation is far beyond multi-scale depth-conv.

---

> > ### Author Response · Authors · 2022-08-02
> > **Response to reviewer EqAk continued**
> >
> > **Q3. Comparison with ConvNext and MixFormer**
> >
> > **Compare with ConvNeXt**. Due to the limited space, we did not put the comparison with ConvNext in our main submission, but we did compare with it in our Supplementary across different model sizes and tasks as shown in Table 20 in our supplementary document. For convenience, we post the numbers here:
> >
> > * Image Classification with multi-scale architecture
> >
> >  Model | Tiny | Small | Base | Large
> > | -- | -- | -- | -- | --
> > | ConvNeXt  | 82.1 | 83.1 | 83.8 | **86.6**
> > | FocalNet    | **82.3** | **83.5** | **83.9** | 86.5
> >
> > * Image Classification with monolithic architecture
> >
> >  Model | Small | Base
> > | -- | -- | --
> > | ConvNeXt  | 79.7 | 82.0
> > | FocalNet    | **80.9** | **82.4**
> >
> > * Object detection with Mask R-CNN 3x
> >
> >  Model | $AP^b$ | $AP^m$
> > | -- | -- | --
> > | ConvNeXt  | 46.2 | 41.7
> > | FocalNet    | **47.6** | **42.6**
> >
> > * Object detection with Cascade Mask R-CNN 3x
> >
> >  Model | $AP^b$ | $AP^b_{50}$ | $AP^b_{75}$
> > | -- | -- | -- | --
> > | ConvNeXt  | 50.4 | 69.1 | 54.8
> > | FocalNet    | **51.5** | **70.1** | **55.8**
> >
> > * Image Segmentation on ADE20K with UperNet
> >
> >  Model | Tiny | Small | Base
> > | -- | -- | -- | --
> > | ConvNeXt  | 46.7 | 49.6 | 49.9
> > | FocalNet    | **47.2** | **50.1** | **51.1**
> >
> > As we can see, our FocalNet outperforms ConvNeXt in almost all cases. Compared with ConvNeXt, FocalNet also uses depth-wise conv but further develops a focal modulation mechanism, which explains the superiority of our method.
> >
> > **Compare with MixFormer**.  We thank the reviewer for pointing out MixFormer. Since MixFormer used deeper and thinner model architecture (as listed in their Table 2), and we demonstrated that it is better than the shallower and wider layouts as in Swin. For fair comparison, we compare with MixFormer using our thinner and deeper FocalNets on three vision tasks, image classification, object detection, and segmentation.
> >
> > * Image Classification
> >
> >  Model | #Params (M) | FLOPs (G) | Top-1
> > | -- | -- | -- | --
> > | MixFormer-B4 | 35.0 | 3.6 | 83.0
> > | MixFormer-B5 | 62.0 | 6.8 | 83.5
> > | MixFormer-B6 | 119.0 | 12.7 | 83.8
> > | FocalNet-T      | 25.1 | 4.0 | 82.7
> > | FocalNet-S      | 38.3 | 6.4 | 83.5
> > | FocalNet-B      | 85.1 | 14.2 | **84.1**
> >
> > * Object detection with Mask R-CNN
> >
> >  Model | #Params (M) | FLOPs (G) | $AP^b$ ($1\times$) | $AP^m$ ($1\times$)  | $AP^b$ ($3\times$) | $AP^m$ ($3\times$)
> > | -- | -- | -- | -- | -- | -- | --
> > MixFormer-B4 | 53 | 243 | 45.1 | 41.2 | 47.6 | 43.0
> > FocalNet-T | 46 | 261 | **46.8** | **41.9** | **48.5** | **43.3**
> >
> > * Semantic Segmentation with UperNet
> >
> >  Model | #Params (M) | FLOPs (G) | mIoU (single-scale) | mIoU (multi-scale)
> > | -- | -- | -- | -- | --
> > MixFormer-B4 | 63 | 918 | 46.8 | 48.0
> > FocalNet-T      | 55 | 934 | **47.4** | **48.5**
> >
> > The above comparisons demonstrate the superiority of our proposed FocalNets to MixFormer across different tasks. We will add these comparisons in our revision.
> >
> > **Q4. Hierarchical v.s. Parallel contextualization**
> >
> > Thanks for pointing out this. As we discussed in lines 152-156, using hierarchical contextualization can rapidly increase the receptive field by stacking a series of depth-wise convolution layers with small kernel sizes. For example, stacking three convolution layers with kernel sizes 3,5,7 can attain an effective receptive size of 13x13, whereas the parallel contextualization using the same kernel sizes can only capture a 7x7 receptive field maximally. As we showed in our experiments by comparing FocalNet (SRF) and FocalNet (LRF) on image classification, object detection, and segmentation, a larger receptive field can capture longer-range visual dependencies and lead to better performance. For further comparison, we replace the hierarchical contextualization with parallel ones in our FocalNet-T (kernel sizes=3,5,7) without any other changes. This variant performs depth-wise conv on top of the input feature map $Z^0$, and has exactly the same number of parameters and FLOPs. However, due to the decreased effective receptive field, its top-1 accuracy on ImageNet-1K drops to 82.0% from the original 82.3%. When applied this pre-trained model to Mask R-CNN for object detection, the performance drops from 46.1 to 45.7, and 48.0 to 47.7 for 1x and 3x schedules, respectively. Based on these experiments, we recommend using hierarchical contextualization rather than parallel contextualization to capture both short- and long-range contexts more effectively, especially for dense prediction tasks.

---

> > > ### Comment · Reviewer_EqAk · 2022-08-07
> > > **Thanks to authors' response.**
> > >
> > > Thanks to authors' response. Most of my concerns are well addressed. I would like to recommand accept.

---

### Author Response · Authors · 2022-08-02
**Response to all reviewers**

First of all, we sincerely thank all reviewers for their constructive comments and valuable suggestions.

We are pleased that all reviewers think our paper is well written and the method clearly presented. We are encouraged that reviewers find our proposed focal modulation interesting and novel (EqAk, RTBL), recognize the thoroughness of our experiments and analysis (EqAk, RTBL, pZvC, 1dzg) which renders promising SoTA performance (EqAk, pZvC).

We carefully read the comments from each reviewer and attempted to provide comprehensive responses accordingly. Please find the separate response below each official review. We hope the responses could answer the questions raised by reviewers and address any concerns about our work.

Thanks again to all reviewers for the time and effort!

---

### Meta-Review · Area_Chair_nu1z · 2022-08-24

**Recommendation:** Accept
**Confidence:** Certain

**Metareview:**

All the reviewers acknowledge that the paper is well-written, novel, and shows strong performance gain. Besides, all the reviewers are satisfied with the authors' response to the raised concerns. AC double-checks the paper, reviews, and response, and finds that the paper is well-shaped and generally flawless. AC recommends acceptance.

**Award:**

No

---

### Decision · Program_Chairs · 2022-09-14

Accept